# Limitation of amino acid availability by bacterial populations during enhanced colitis in IBD mouse model

Tanner G. Richie,[1] Leah Heeren,[1] Abigail Kamke,[1] Kourtney Monk,[1] Sophia Pogranichniy,[1] Trey Summers,[1] Hallie Wiechman,[1] Qinghong Ran,[1] Soumyadev Sarkar,[1] Brandon L. Plattner,[2] Sonny T. M. Lee[1]

**ABSTRACT** Members of the Enterobacteriaceae and *Enterococcus* are associated with persistent gut inflammation due to rapid colonization combined with pathogenic tendencies. Here, we investigated the functions of gut microbial populations resulting in persistent gut inflammation. In this study, we utilized the IL-10 knockout mouse model and induced colitis using dextran sulfate sodium (2%) after development. Dams during gestation were provided cefoperazone to induce vertically transmitted dysbiosis in the pups that were monitored in this study. We characterized the dysbiotic gut microbial community and potential crosstalk of these microbes, and host gene expression changes to identify bacterial populations and potential functions that were involved in gut inflammation. We isolated Enterobacteriaceae populations from mice to validate the utilization of sulfur-containing amino acids. Members of Enterobacteriaceae and *Enterococcus* were highly detected in inflamed mice. Enterobacteriaceae populations containing L-cysteine dioxygenase were strongly correlated with the upregulation of host gene *CSAD*, responsible for cysteine breakdown. We observed that bacterial isolates from dysbiotic mice displayed increased growth rates when supplemented with L-cysteine, highlighting the use of sulfur metabolism. Our results show that microbial populations use alternate metabolisms and sequester host nutrients for growth, associated with inflammation in the gut.

**IMPORTANCE** Inflammatory bowel disease is associated with an increase in Enterobacteriaceae and Enterococcus species; however, the specific mechanisms are unclear. Previous research has reported the associations between microbiota and inflammation, here we investigate potential pathways that specific bacteria populations use to drive gut inflammation. Richie et al. show that these bacterial populations utilize an alternate sulfur metabolism and are tolerant of host-derived immune-response products. These metabolic pathways drive host gut inflammation and fuel over colonization of these pathobionts in the dysbiotic colon. Cultured isolates from dysbiotic mice indicated faster growth supplemented with L-cysteine, showing these microbes can utilize essential host nutrients.

**KEYWORDS** fecal microbiota transplantation, inflammatory bowel disease, microbial-host interactions, amino acid uptake, RNA sequencing, shotgun metagenomics

Inflammatory bowel disease (IBD) is a category of chronic immune-mediated disorders with a wide range of outcomes and an unresolved etiology. The incidence of IBD is increasing globally, with over 6 million cases, and a significantly increased number of diagnoses reported within the last three decades (1). The tri-factor of genetic, environmental, and microbial components is suspected to each play a significant role in the development of these complex disorders (2–4). A microbial cause for IBD has been suspected as far back as the mid-19th century when the disease was first described

Address correspondence to Sonny T. M. Lee, leet1@ksu.edu.

The authors declare no conflict of interest.

See the funding table on p. 18.

by Samuel Wilks in 1859 (5). While gut dysbiosis is associated with IBD, it has been difficult to definitively determine whether these changes observed are causal or merely a consequence of IBD. It is possible that dysbiosis initiates or contributes to the progression of IBD by amplifying and sustaining the local or systemic inflammatory process (6, 7).

There is no consensus that IBD is caused by classical pathogens. Several studies show that commensal microbes can display pathogenic properties given the right circumstances, context, and opportunity (6, 8–10). These microbes have been called "pathobionts" (8, 11). Specific microbes have been hypothesized to play a role in the development or aggravation of IBD (12). Adherent invasive *Escherichia coli* (AIEC) and *Enterococcus faecalis* have been observed in patients with IBD and Crohn's disease (CD) who display virulence factors at higher rates compared to healthy patients (9, 12). However, we still have a limited understanding of the interactions between these bacteria and the host during the onset of IBD and related gastrointestinal inflammatory disorders (13, 14). In addition to the identification of relevant potential causal microbes of IBD, the function of these microbes before and during the disease, and how they interact with the host are crucial in understanding this complex disease dynamic (15, 16). For example, *Akkermansia muciniphila* and butyrate-producing *Clostridia* have been shown to improve gut physiology and even alleviate symptoms of IBD and inflammation in both mice and humans (17, 18). On the other hand, AIEC has been proposed as a possible cause of IBD, and studies have also found an increased number of virulent *E. coli* strains isolated from IBD patients (12, 19). Some studies suggest that the accumulation of AIEC and other *E. coli* strains in the gut is a consequence of inflammation, while other studies show that AIEC infection induces changes in the gut microbiota (20–22). It is speculated that certain metabolic pathways of AIEC, such as utilization of host-derived nitrate and uptake of sulfur-rich amino acids like cysteine and taurine (23–25), drive chronic inflammation within the gut environment. Currently, the mechanisms of inflammation associated with bacterial populations in the dysbiotic remain unclear.

In patients with IBD, the chronically inflamed small intestine and colon are subjected to considerable oxidative stress (26, 27). Amino acid metabolism plays an important role in the nutrition and physiology of the host and may influence the progress of IBD (25, 28). Studies demonstrated that amino acids are critical for mucosal wound healing and are used as energy substrates of enterocytes (15, 29, 30). These amino acids may act to reduce inflammation, oxidative stress, and proinflammatory cytokines. Although experimental studies investigating the therapeutic efficacy of amino acids are promising, some clinical studies with supplementation in IBD patients are disappointing (29, 31, 32), lacking high-resolution data in both host and microbial responses. Conflicting results warrant additional research with the goal of a deeper understanding of how amino acid metabolism affects the underlying mechanisms of IBD.

In this study, we sequenced and assembled high-quality genomes of bacterial populations for system-wide simultaneous analysis of host gene expression and microbial functions. To focus on novel host-microbial interactions that might reveal microbial impacts on the pathogenesis of IBD, we analyzed microbial functions and metabolic pathways that were complemented by the host. Using a combination of bioinformatic, genomic, and biochemical approaches, we highlighted biosynthesis and degradation pathways of cysteine and taurine in Enterobacteriaceae and *Enterococcus* populations that might contribute to the initiation, amplification, and sustaining intestinal inflammation in long term, vertically transmitted dysbiosis (phosphate buffered saline ]PBS] Gavage and No Gavage groups) IL-10 KO mice. Protection from inflammation and colon damage was demonstrated in control and fecal microbiota transplantation (FMT) groups with lower histologic inflammation scores and several immunologic indicators. Importantly, we also showed that Enterobacteriaceae populations isolated from mice displayed growth specifically by utilization of L-cysteine. Our experiments advance our fundamental understanding of the impact of host-microbe interaction on intestinal inflammation and showed that the understanding of the

congruent expressions of host gene expressions and microbial functions represents a robust way to discover potential implications of pathobionts, and thereby provides a target for the generation of novel microbial treatment options for IBD.

## MATERIALS AND METHODS

### Sample collection and processing

#### IL-10 KO mice

All mice in this study were IL10 KO Jax stock #002251/C57BL/6J genetic background. Mice were bred in the Kansas State University Johnson Cancer Research Center mouse facility under *Helicobacter hepaticus*-free conditions with regular chow, autoclaved drinking water, and housed up to four littermates of the same sex per cage. Here, we use one of the most common IBD murine models, the IL-10 KO mouse with colitis induced by a common chemical colitogen, dextran sulfate sodium (DSS) (33). These mice lack IL-10, a regulator of the immune response leading to the development of colitis later in life (34). Figure 1 shows the experimental setup and treatment protocol of the study. To investigate the impact of the microbial populations on inherited dysbiosis, we conducted using four different experiment sets of mice—control mice (Control); mice (with inherited dysbiosis) that were gavaged with control mice microbiota (FMT); mice (with inherited dysbiosis) that were gavaged with PBS (PBS); and mice (with inherited

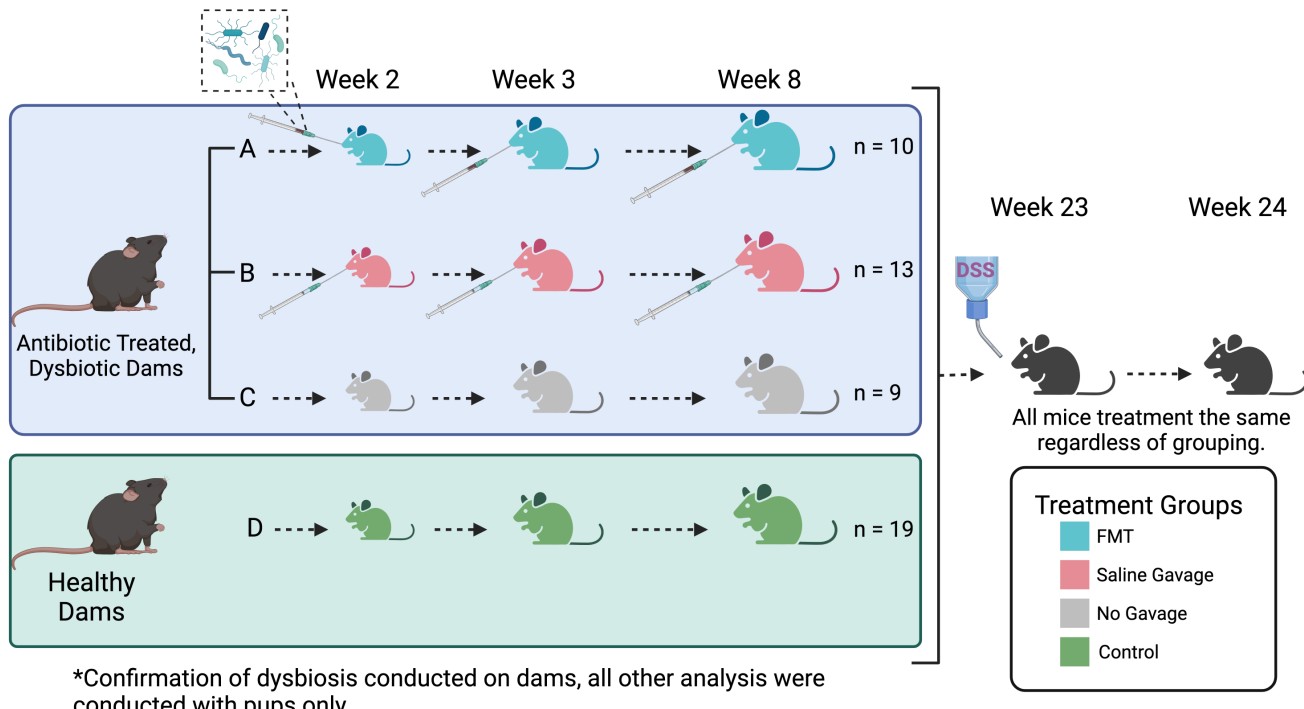

**FIG 1** Graphic of experimental design highlighting experimental conditions and the timeline by the inclusion of important dates and defining the treatment groups. Briefly, Il-10 KO Dams' drinking water was supplemented with cefoperazone sodium salt to instill dysbiosis, which was inherited in all pups except for control pups. Dysbiotic-inherited and control pups were then treated with 2% DSS at week 23 of age to chemically induce IBD.

dysbiosis) that did not receive any gavage (No gavage). To instill inherited dysbiosis (35), we supplemented dams with autoclaved drinking water with cefoperazone sodium salt (CPZ; 0.5 mg/mL), a broad-spectrum antibiotic, when they were in the third week of gestation. The duration of CPZ exposure lasted until pups were weaned at 21 days of age (Fig. 1). We collected fecal samples from control mice daily and were frozen at −80 °C. To prepare the slurry for gavaging, we thawed, pooled, and gently vortexed control fecal samples in sterile PBS, then immediately frozen to −80°C until use. We gavaged the mice either with control mice slurry (FMT) or PBS (PBS) at 2, 3, and 8 weeks of age. Mice were monitored for signs of colitis. By week 23, the remaining mice received 2% DSS for 5 days in autoclaved drinking water and were changed as needed. We monitored mice daily during the time of DSS treatment and after treatment, until the end of the study. During a 16-day recovery period after 2% DSS treatment, all mice were sacrificed, and tissues were collected.

## Serum

We collected blood via heart puncture and centrifuged it to collect serum. The serum was stored at −80°C until further analysis. Mouse cytokines were obtained using a 31-Plex assay (Eve Technologies, Canada, catalog: #33836) that measured eotaxin, granulocyte colony-stimulating factor (G-CSF), granulocyte-macrophage (GM)-CSF, interferon-gamma (IFNγ), interleukin (IL)-1α, IL-1β, IL-2, IL-3, IL-4, IL-5, IL-6, IL-7, IL-9, IL-10, IL-12 (p40), IL-12 (p70), IL-13, IL-15, IL-17A, IP-10, KC, LIF, LIX, MCP-1, M-CSF, MIG, MIP-1α, MIP-1β, MIP-2, RANTES, TNFα, and VEGF-A (not validated).

## Histopathology

A representative section of the lower colon from each mouse was routinely fixed in 10% neutral-buffered formalin, trimmed, embedded in paraffin, sectioned, and then stained with hematoxylin and eosin. Tissues were blindly scored for characteristics of inflammation including surface epithelial damage, type and number of infiltrating inflammatory cells, and gland damage including dilation, inflammation, and hyperplasia. Each category was graded on a scale from 0 (normal), 1 (mild), 2 (moderate), to 3 (severe). Summative scores for each mouse and means for each treatment group were calculated.

## Shotgun metagenomics

We collected the fecal samples ($n$ = 51) and immediately froze to −80°C until extraction was performed. We extracted the gDNA using the E.Z.N.A Stool DNA Kit, following the manufacturer's protocol for pathogen detection supplemented with 10 minutes of bead beating. Extracted DNA was kept at −20°C until shotgun metagenomic sequencing. Sample DNA concentration and quality were determined by Qubit and Nanodrop. The products were then visualized on an Agilent Tapestation 4200 and size-selected using the BluePippin. DNA libraries were constructed using Illumina's Nextera XT Index Kit v2 kit. The final library pool of 51 samples was quantified on the Kapa Biosystems qPCR protocol and sequenced on the Illumina NovaSeq S1 chip in a 2 × 150 bp paired-end sequencing run. A negative-control sample was also included in the process to ensure there was no contamination from molecular DNA extraction and sequencing.

## Colon RNA sequencing

A portion of the lower large intestine was stored in TRIzol reagent (−80°C) until sequencing. Samples were sent to CD Genomics (Shirley, New York), where they were extracted, and QC analysis was performed by manufacturer protocols. Next, cDNA library preparation (poly-A selection) was completed for the 12 samples (three per treatment group) and were then sequenced via NovaSeq PE150 for a total of 20 million pair-end reads.

## Bacterial culturing and nutrient dependency assay

Enteric bacteria from −80°C frozen mouse fecal samples from each treatment group were isolated on MacConkey agar. Fecal pellets were vigorously homogenized in sterile PBS and streaking serial dilutions up to $10^{-5}$, incubating at 37°C overnight. Isolation streaks were performed twice for each isolate and DNA was extracted using the E.Z.N.A Stool DNA Kit. Extracted DNA was sent out for Sanger sequencing by Functional Biosciences (Madison, Wisconsin). The bacterial growth curves were performed using M9 broth (Sigma) supplemented with magnesium chloride, calcium chloride, and 20% glucose. Media was supplemented with 100 uM of cysteine and growth was monitored every 4 hours for 12 hours by $OD_{600}$ measurements.

## Bioinformatic workflow

### Metagenomics

We automated our metagenomics bioinformatics workflows using the program "anvi-run-workflow" in anvi'o ver 7.1 (36). The workflows use Snakemake v6.15.5 (37) and implement numerous tasks detailed in the following sections, including short-read quality filtering, assembly, gene calling, functional annotation, hidden Markov model (HMM) v 11.0 search (38), metagenomic read-recruitment and binning (39).

We used the program "iu-filter-quality-minoche" to process the short metagenomic reads, and removed low-quality reads according to the criteria outlined by Minoche et al. We used MEGAHIT v1.2.9 (40) to co-assemble quality-filtered short reads into longer contiguous sequences (contigs). We assigned the metagenomes into four experimental groups (control/no-CPZ, $n = 19$; CPZ/no-gavage, $n = 9$; CPZ/gavage-with-PBS, $n = 13$; CPZ/FMT, $n = 10$) for the co-assembly. We then used the following strategies to process the assemblies: (i) "anvi-gen-contigs-database" on contigs to compute k-mer frequencies and identify open reading frames (ORFs) using Prodigal v2.6.3 (41); (ii) "anvi-run-hmms" to identify sets of bacterial and archaeal single-copy core genes using HMMER v.3.2.1 (42); (iii) "anvi-run-ncbi-cogs" to annotate ORFs from NCBI's Clusters of Orthologous Groups (COGs) version 2014 (43); and (iv) "anvi-run-kegg-kofams" to annotate ORFs from KOfam version 2021-10-03 HMM databases of KEGG orthologs version 100.0 (44). We mapped metagenomic short reads to contigs using Bowtie2 v2.3.5 (45) and converted them to BAM files using samtools v1.9 (46). We profiled the BAM files using "anvi-profile" with a minimum contig length of 1,000 bp. We used "anvi-merge" to combine all profiles into an anvi'o merged profile for all downstream analyses. We then used "anvi-cluster-contigs" to group contigs into initial bins using CONCOCT v1.1.0 (47), and used "anvi-refine" to manually curate the bins based on tetranucleotide frequency and different coverage across the samples. We marked bins that were more than 70% complete and less than 10% redundant as metagenome-assembled genomes (MAGs) (48). Finally, we used "anvi-compute-genome-similarity" to calculate the average nucleotide identity (ANI) of our genomes using PyANI v0.2.9 (49) and identified non-redundant MAGs. We conducted all downstream analyses on only non-redundant MAGs. We used the "detection" metric to assess the occurrence of MAGs in a given sample. Detection of the MAGs is the proportion of the nucleotides that were covered by at least one short read. We considered a MAG was detected in a metagenome if the detection value was >0.25, which is an appropriate cutoff to eliminate false-positive signals in read recruitment results, for its genome.

### PATRIC analyses

Non-redundant MAGs were uploaded and annotated in the Pathosystems Resource Integration Center (PATRIC) (50). For the MAGs of interest, similar genomes were identified, and phylogenetic trees were built using 50 similar strains to identify close relationships and approximate identity. Next, for each MAG, Subsystem genes were utilized for the comparison of potential functions and virulence factors between the MAGs of interest. Counts of genes by each class of interest were used to build a heatmap

and pathway constructs for each MAG of interest using the Pathway function in PATRIC linked to KEGG.

## RNASeq analysis

We used FastQC (51) and multiQC (52) to check the raw reads' quality and used Trimmomatic (53) to trim the reads. We used Trinity ver 2.13.1 (54) for *de novo* assembly, and RSEM (55) and DESeq2 (56) to estimate the expression levels and differential expression analysis, respectively. Annotation was carried out using blast+ 2.11 using default parameters and arguments (57) and R version 4.0.0. The transcripts compared in DESeq2 were first blasted against the mouse reference genome C57BL/6NJ as a reference ensemble ID (http://ftp.ensembl.org/pub/release-107/fasta/mus_muscu-lus_c57bl6nj/cdna/). Then the top hits (lowest e-value and largest length with a minimum of 150 bp) were kept, in addition a *P*-value cutoff of 0.01 using the default Benjamini and Hochberg method, and a fold change cutoff of 2 was implemented before downstream analysis. Finally, we used GO_MWU (58) to analyze the annotated transcripts.

## Statistical analyses

Differences between groups were analyzed by one-way ANOVA with pairwise comparisons and Tukey post hoc test unless stated otherwise. *P*-values of less than 0.05 were considered statistically significant, with standard error of the mean (mean ± SE) utilized where appropriate. Statistical analyses were performed in RStudio. We used the program—Data Integration Analysis for Biomarker Discovery using Latent cOmponents (DIABLO) to build a model to identify key components in the metagenomic data and host RNASeq data (59, 60). We (i) first identified key omics variables in the integration process; (ii) and then maximized the common and correlated information between the data sets; and (iii) finally visualized the results to identify relevant microbial detections and host genomic markers.

## RESULTS

Briefly, IL-10 KO mice were utilized for this study, pregnant dams were given CPZ in drinking water until pups were weaned (61). These pups, with vertically transmitted dysbiosis, were used for all downstream analysis and at the endpoint of this study (week 24) colitis was induced using 2% DSS; detailed experimental breakdown can be found in the Section Materials and Methods. Fecal content from pups was collected at the final week (week 24) of the study (Fig. 1). Genomic DNA was extracted, and shotgun sequencing was conducted. Mouse pups displayed vertically transmitted dysbiosis from CPZ antibiotic exposure of the dams, except for the control group (control/no-CPZ), and received FMT at weeks 2, 3, and 8 (CPZ/FMT), saline gavage at weeks 2, 3, and 8 (CPZ/gavage-with-PBS), or no gavage (CPZ/no-gavage) during and after development (Fig. S1). We employed genome-resolved metagenomics (62) for an in-depth characterization of the gut microbiota and combined with the host response markers—colon RNA sequencing, serum immune markers, and histology to provide insights into the pathogenesis of the microbiota.

Shotgun sequencing of the fecal samples resulted in a total of over 623 million sequences with an average of 12.2 ± 1.6 (mean ± SE) million reads per metagenome. Co-assemblies of the four experimental groups ($n$ = 51; control/no-CPZ, $n$ = 19; CPZ/no-gavage, $n$ = 9; CPZ/gavage-with-PBS, $n$ = 13; CPZ/FMT, $n$ = 10) metagenomes resulted in an average of 90 ± 14.3 thousand contigs that were longer than 1,000 nucleotides.

Dysbiosis was confirmed using detection value shifts of MAGs resolving to Firmicutes across each mouse pup and corresponding dam. Dam fecal samples were taken before and after exposure to the antibiotic cefoperazone, to capture the modulation in gut microbiota makeup due to antibiotic exposure. Our data clearly indicated a large shift in detection levels of most Firmicutes, indicating gut dysbiosis (63–65). Pup samples

clustered near antibiotic-treated dams reveal high similarity between detection patterns, indicating vertically transmitted dysbiosis (Fig. S1). All following data analyses and experiments in this study were conducted on the IL-10 knockout pups' development to determine the combined impact of vertically transmitted dysbiosis (drug exposure components of IBD) and gut microbiota (microbial component of IBD).

## FMT resulted in reduced colon inflammation and changes in systemic serum markers

We investigated the host response to FMT to gain a perspective of the impacts later in life from reintroducing native microbes during the development of the neonates. We noticed that mice that received FMT displayed less colon surface damage, fewer lamina propria infiltrating lymphocytes, and significantly lower inflammation scores than dysbiotic mice that did not receive FMT ($P < 0.015$). Mice that received PBS gavage or no gavage displayed the highest histopathologic score with evidence of epithelial damage, lymphocyte infiltration, and loss of goblet cells ($P < 0.009$) (Fig. 2A; Table S2). Select systemic serum markers indicative of broad-scale inflammation were measured in the serum of IL-10 KO mice. The cytokine panel showed low-level trends for the control mice compared to other groups (Fig. 2B; Table S2). Notably, FMT mice resulted in significantly higher levels of G-CSF ($P = 0.00016$), and IL-6 ($P = 0.0019$) compared with control mice, suggesting that certain beneficial microbes, when acquired early in life, play a deterministic role in host immune modulation and priming, as well as maintaining homeostasis (66, 67). Some inflammatory cytokines were significantly higher in the No Gavage compared to FMT and control groups, respectively, IFNγ ($P = 0.0063$) and IL-17 ($P = 0.039$), suggesting these mice may experience higher inflammatory responses in the gut. With few significant increases in common IBD inflammation indicators like IL-1β and only No Gavage mice displaying increased IL-17 (68, 69), we observed little evidence of a systemic, uncontrolled, immune response, regardless of treatment group. In summary, IL-10 KO mice receiving FMT displayed less colonic inflammation with little change in host-associated cytokine modulation compared with other groups, indicating an overall successful FMT.

## Upregulation of several IBD and colon cancer-associated genes in dysbiotic mice highlights the importance of microbial biodiversity

To further investigate host response to long-term dysbiosis and the impact of microbial intervention, we performed RNA sequencing on mice colon samples ($n = 3$) from each treatment group. First, we performed a gene ontology (GO) analysis to gain insight into broad gene categories that were disproportionately upregulated or downregulated by treatment group comparisons. Six pairwise comparisons revealed gene regulation changes between treatment groups (Fig. S2). Comparisons between control and FMT mice showed only six gene categories significantly different (three upregulated and three downregulated classes), while significantly more differences were seen with comparisons between FMT and the CPZ/gavage-with-PBS (35 upregulated and 32 downregulated; 67 total), as well as FMT compared with CPZ/no-gavage (84 upregulated and 45 downregulated; 129 total). Similarly, comparisons between control mice and CPZ/gavage-with-PBS (20 upregulated and 34 downregulated; 54 total), and the CPZ/no-gavage groups (59 upregulated and 29 down; 88 total) also showed several categorical differences. We observed significant differences in several gene families between dysbiotic and FMT mice groups, including multiple inflammatory cytokine/chemokine activity, sulfur dioxygenase, sulfur transport activity, cysteine receptor binding, and nitric oxide synthase binding. Nearly identical trends in gene family differences between control mice and dysbiotic groups were observed. Finally, differences between FMT and control groups were fewer, with significant differences noted in gap junction channel activity, cytokine/chemokine activity, IL-17 receptor activity, and sulfur compound binding, all suggesting the FMT successfully reduced gut inflammation in the host. We examined these gene families because of their known involvement in

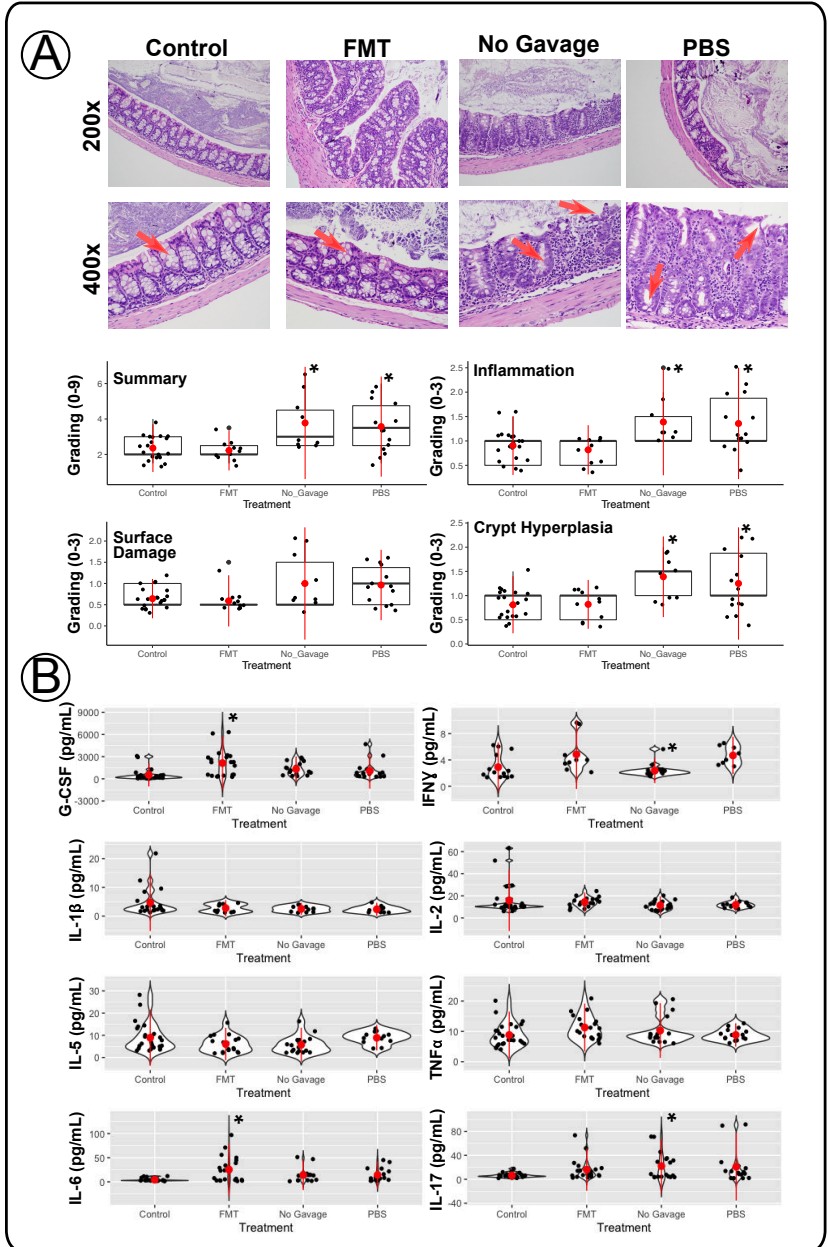

**FIG 2** Dysbiosis drives persistent host response in colon inflammation with the absence of a large systemic response. (A) Histologic images of IL-10 KO colon representing each treatment group's average score. Flags indicate sites of goblet cell status, highlighting more in control/FMT groups compared to dysbiotic groups or evidence of surface damage as indicated. Boxplots of the blinded scores of each treatment group are included. Summary scores (0–9) (total of each category) are shown for each treatment group in addition to individual scores for inflammation (0–3), surface damage (0–3), and crypt hyperplasia (0–3) for each mouse treatment group shown on the *y*-axis. Summary scores showed reduced colon inflammation in FMT treatment pups. (B) Violin plots of pro-inflammatory cytokine measurements of IL-10 KO serum for cytokines relevant to IBD development showed that FMT resulted in systemic bio-immune markers. Red bars on graphs mark the average with standard deviation for each treatment group. One-way ANOVA and pairwise comparisons with Tukey adjustments were used for all results in this figure (* indicates *P* < 0.05).

the development or more severe symptoms of IBD in human patients (25, 70, 71). As discussed previously, cytokine and chemokine modulators initiate inflammation during an immune response, with IFNγ, IL-6, and IL-17 acting as early indicators of IBD (69). Sulfur dioxygenase activity is a precursor step in the production of cysteine, along with the transporter genes to gain access to more sulfur ions to construct cysteine (72). In IBD patients, low cysteine and taurine levels have been observed, with enteric bacteria such as *Escherichia coli* likely benefiting from sulfur metabolism (24, 25). Combining this knowledge with our data, our study indicates that dysbiotic mice might have lower cysteine levels contributing to the upregulation of cysteine biosynthesis gene families. Nitric oxide is also a crucial component of the host immune response to potential pathogens. Nitric oxide, generally toxic to gut microbes (73), is known to be significantly increased in dysbiotic mice, which suggests that its presence indicates a pro-inflammatory immune response in the gut, and is thus another potential biomarker of active IBD (74).

To gain more insights into the host-expressed gene functions, we examined counts of individual genes in treatment group comparisons (Fig. 3A; Table S3). We observed several upregulated genes associated with colitis and IBD development, including *ABCA2*, *Acss1*, *Araf*, *CSAD*, *Ilf3*, and *TRAFD1,* showing similar fold change patterns across treatment groups (Fig. 3B). Each of these genes were upregulated in dysbiotic mice as compared to the CPZ/FMT and control/no-CPZ groups. These genes function as an ABC transporter upregulated in several cancers (*ABCA2*)( 75 ), a member of the tricarboxylic acid cycle involved in metabolism of tumorigenesis (*Acss1*) (76), growth promoter kinase, a proto-oncogene present in intestinal cancers(*Araf*) (77), cysteine desulfurase, shown upregulated in Crohn's disease (*CSAD*) ( 78), interleukin binding enhancer (*Ilf3*) (79), and a negative regulator of the innate immune response against pathogens (*TRAFD1*) ( 80) that are found to be upregulated in IBD and colon cancer development. *TRAFD1* upregulation suggests a host attempt to slow immune response common during pathogen infection response. *CSAD* gene upregulation seen in the dysbiotic groups indicated that the host was increasingly utilizing cysteine as a sulfur source; a common finding in IBD patients is low serum cysteine levels (28), which could indicate the dysbiotic mice in our study were experiencing low cysteine levels as well. Formation of homocysteine occurs during

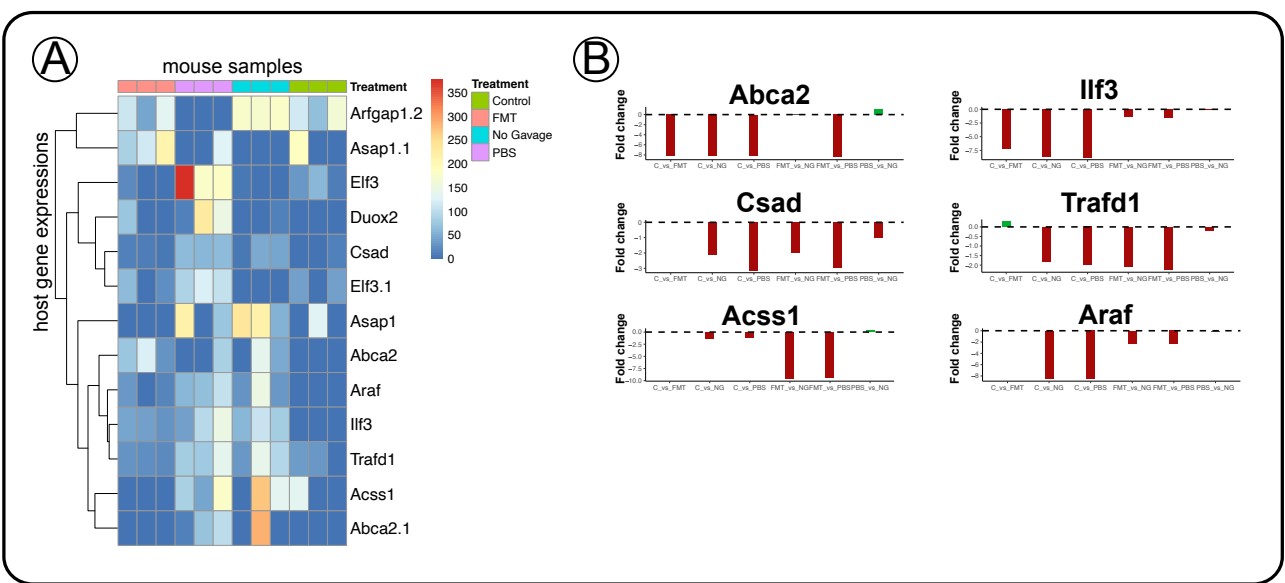

**FIG 3** IBD and cancer-associated genes upregulated in dysbiotic IL-10 KO mice, showing the success of pups that were treated with FMT as compared with control mice. (A) Heatmap of significantly regulated gene counts across treatment groups, including 3 samples per group. (B) Fold change plots of selected significantly different regulated genes by treatment group comparisons. The full set of gene counts and significantly regulated genes are in Table S6.

biosynthesis of cysteine, and an increase in homocysteine levels has been shown in IBD patients, indicating that the host might be starved of cysteine (81, 82). The genes accounted for here were also represented above in the gene family categories that were differentially regulated in dysbiotic mice compared to FMT and control mice. Our results collectively suggest that these genes are an indication of a host's response to dysbiosis in the gut, and its immune responses to regulate inflammation.

## Distinct bacterial populations are associated with inherently dysbiotic IL-10 KO pups

Due to changes in certain bacterial populations, conditions of dysbiosis-associated gut are reported to result in chronic persistent intestinal inflammation (19, 83). Therefore, with the knowledge of the hosts' gene expressions, we next examined the effect of reintroducing native intestinal microbial communities via FMT early in life on gut microbial composition post-development. We reconstructed MAGs that were >70% complete with <10% redundancy as predicted by bacterial single-copy core genes as previously described (84, 85), from the co-assembled treatment group metagenomes (86, 87). We resolved a total of 677 MAGs, and then removed redundancy by selecting a single representative for each set of genomes, resulting in 190 non-redundant MAGs (Table S4) from the four experimental groups. There was a large representation of Firmicutes ($n = 178$) with Proteobacteria ($n = 6$), Bacteroidota ($n = 1$), Actinobacteria ($n = 3$), and Verrucomicrobiota ($n = 2$) having lower detection. At the class level, Clostridia ($n = 158$) was the most dominant, followed by Bacilli ($n = 17$), Gammaproteobacteria ($n = 3$), Coriobacteriia ($n = 3$), Negativicutes ($n = 1$), Dehalobacteriia ($n = 1$), Bacteroidia ($n = 1$), Verrucomicrobia ($n = 1$), and five remained unidentified (Fig. 4A).

We observed that the gut microbiota of mice that received FMT resembled that of mice that were not dysbiotic. Dysbiotic mice that received either gavage of PBS or no gavage were distinctly different from non-dysbiotic and FMT mice (Fig. 4A; Table S4). We used the MAG's detection values to calculate the Bray-Curtis dissimilarity and showed that there was a significant clustering of control/no-CPZ and CPZ/FMT group (Fig. 4A). We also observed a wider clustering pattern in the CPZ/no-gavage and CPZ/gavage-with-PBS groups. The differences among the treatment groups were mainly attributed to the absence and presence of the families Enterococcaceae and Enterobacteriaceae, which exhibited the largest differences in MAG's detection (Table S4). Our results were corroborated by other studies that also showed an increase in the relative abundance of bacterial populations from families Enterococcaceae and Enterobacteriaceae (12, 19).

Although previous studies have identified changes in the relative abundance of Enterococcaceae and Enterobacteriaceae bacterial populations, it has been challenging to identify the exact culprit causing gut inflammation due to the resolution of the studies (88, 89). In our study, of the 190 non-redundant MAGs, we resolved to a high-resolution level, 7 MAGs that were of interest—MAGs 001 and 002 (hereafter referred as MAGs-*Akkermansia*), MAGs 165 and 168 (hereafter referred as MAGs-*Enterococcus*), and MAGs 166, 167, and 169 (hereafter referred as MAGs-Enterobacteriaceae). We resolved MAGs-*Akkermansia* to their respective species level (MAG 001: *Akkermansia muciniphila* and MAG 002: *CAG-485 sp002362485*), while MAGs-*Enterococcus* were also annotated to the species level (MAG 165: *Enterococcus faecium* and MAG168: *Enterococcus faecalis*). While assigning the identity to MAG 166 (Enterobacteriaceae), and MAG 167 (Enterobacteriaceae) was challenging due to the high heterogeneity among the bacterial populations in this family, we managed to resolve both MAGs to a 100% completion (MAG 166—redundancy value 5.6%, and MAG 167—redundancy value 1.4%), obtaining valuable MAG genomic information for downstream analyses. In addition, MAG 169 was assigned to the species *Proteus mirabilis* (completion 100%, redundancy 0%). We used the Pathosystems Resource Integration Center (PATRIC) web portal (50, 90) and constructed phylogenetic trees to further verify the identity of the MAGs. We verified that MAG 001 annotated to *Akkermansia muciniphila*, and MAG 002 was closely related to the *Barnesiella* genus in the Bacteriodota phylum. The other five MAGs were from two different

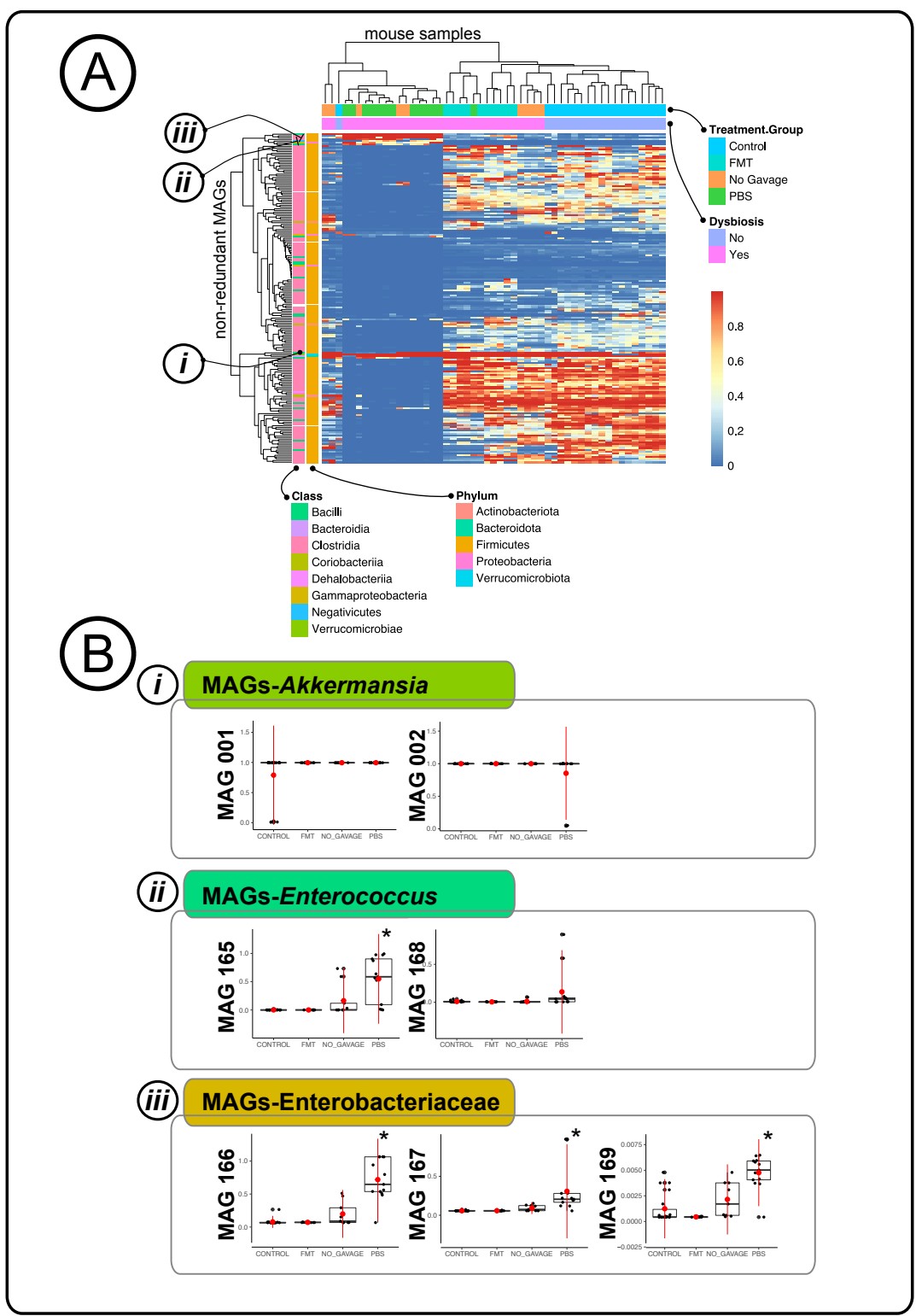

**FIG 4** Distinct bacterial populations that were associated with inherently dysbiotic IL-10 KO pups. (A) Clustered heatmap of IL-10 KO with the non-redundant MAGs, showing clustering of the samples was mainly influenced by the presence and absence of Enterobacteriaceae and *Enterococcus* bacterial populations. (B) Boxplots of detection values of specific MAGs of interest, *Akkermansia*, *Enterococcus*, and Enterobacteriaceae by treatment group, showing the high detection of MAGs-Enterobacteriaceae and MAGs-*Enterococcus* in samples that exhibited symptoms of IBD (* indicates $P < 0.05$).

taxonomic groups—*Enterococcus* (MAG 165 and 168) with the closest relationship to *Enterococcus faecium* and *Enterococcus faecalis,* respectively, and Enterobacteriaceae (MAGs 166, 167, and 169) had the closest phylogenetic relationship with *Enterobacter*, *Kosakonia*, and *Proteus*, respectively (Fig. S3). Out of the three groups of MAGs of interest, MAGs-*Akkermansia* was highly detected in all of the samples (control/no-CPZ: $0.89 \pm 0.0496$; CPZ/no-gavage: $0.99 \pm 0.0005$; CPZ/gavage-with-PBS: $0.92 \pm 0.0505$; CPZ/FMT: $0.998 \pm 0.0005$), regardless of whether the mice were dysbiotic or not. MAGs-*Enterococcus* and MAGs-Enterobacteriaceae had a very different detection pattern as that of MAGs-*Akkermansia* (Fig. 4B). Both MAGs-*Enterococcus* and MAGs-Enterobacteriaceae displayed low detection in the control/no-CPZ (MAGs-*Enterococcus:* $0.005 \pm 0.0015$, MAGs-Enterobacteriaceae: $0.005 \pm 0.0035$) and CPZ/FMT (MAGs-*Enterococcus:* $0.002 \pm 0.0005$, MAGs-Enterobacteriaceae: $0.003 \pm 0.0006$) groups, however, were highly detected in the CPZ/no-gavage (MAGs-*Enterococcus:* $0.08 \pm 0.0501$, MAGs-Enterobacteriaceae: $0.05 \pm 0.0226$) and CPZ/gavage-with-PBS (MAGs-*Enterococcus:* $0.34 \pm 0.0776$, MAGs-Enterobacteriaceae: $0.30 \pm 0.0583$) group. MAGs-*Akkermansia* were highly detected in all treatment groups with no significant differences between any treatment groups. Detections of Enterobacteriaceae MAGs by treatment group indicated a lower presence in control/no-CPZ and mice that receive FMT with no significant differences between these groups; however, significantly higher detections were observed in the dysbiotic mice with the PBS gavage (MAG 166: $P < 0.0001$; MAG 167: $P < 0.011$; MAG 169: $P < 0.0005$), however, not in the no gavage (MAG 166: $P = 0.39$; MAG 167: $P = 0.94$; MAG 169: $P = 0.052$). MAGs-*Enterococcus* showed a similar trend of control mice and FMT mice showing lower detection ratios than PBS gavage (MAG 165: $P < 0.0001$; MAG 168: $P = 0.068$); however, no difference in the no gavage group (MAG 165: $P = 0.42$ MAG 168: $P = 0.99$), respectively (Fig. 4B). We surmised that MAGs-*Enterococcus* and MAGs-Enterobacteriaceae possessed functions that not only enable them to proliferate and outcompete other microbial competitors during dysbiosis but may also have contributed to persistent gut inflammation in the host.

## Coherence between host and microbe highlighted attributes to worsened outcomes in IBD

Identifying host-microbe interactions during dysbiosis-associated intestinal inflammation can provide insight into both the host and microbial pathways that affect inflammation outcomes. So far, we identified specific host modulations in response to FMT, as well as specific bacterial populations that were associated with the dysbiosis. To further determine likely interactions between host and microbe modulations, we employed a discriminant analysis using the mixOmics package encoding DIABLO in R (59). We used a correlation cutoff at 0.85, a stronger correlation cutoff than mixOmics default of 0.7 (91), to ensure robust and biologically relevant linkages between host genes and MAGs (Fig. 5). We demonstrated that Enterobacteriaceae MAGs were positively correlated with inflammatory and host amino acid metabolic response genes, while the Lachnospiraceae and Oscillospiraceae families were negatively correlated with inflammatory genes. Our analysis suggested that MAGs from the Enterobacteriaceae family might drive host response in a dysbiotic gut through crosstalk with the host immune system, and uptake of essential amino acids. We showed that two host genes displayed both positive and negative correlations between a set of MAGs: *CSAD* and *TRAFD1*. *CSAD*, as shown above, to be upregulated in dysbiotic mice, was positively correlated to two MAGs: Enterobacteriaceae MAGs 167 and 169 (Fig. 5). This gene was also negatively correlated with several MAGs closely matching the families Lachnospiraceae and Oscillospiraceae. The gene *CSAD* catalyzes the decarboxylation of cysteine sulfinate to hypotaurine, eventually to taurine (92), with cysteine shown to be essential in the recovery of the host from chronic colon inflammation and reduced IBD symptoms (29, 31). Thus, the strong positive correlation between *CSAD* and MAGs-Enterobacteriaceae suggests that these MAGs may be competing with the host for sulfur-rich amino acids cysteine and taurine as an alternate metabolic pathway to fuel growth leading to over colonization of these

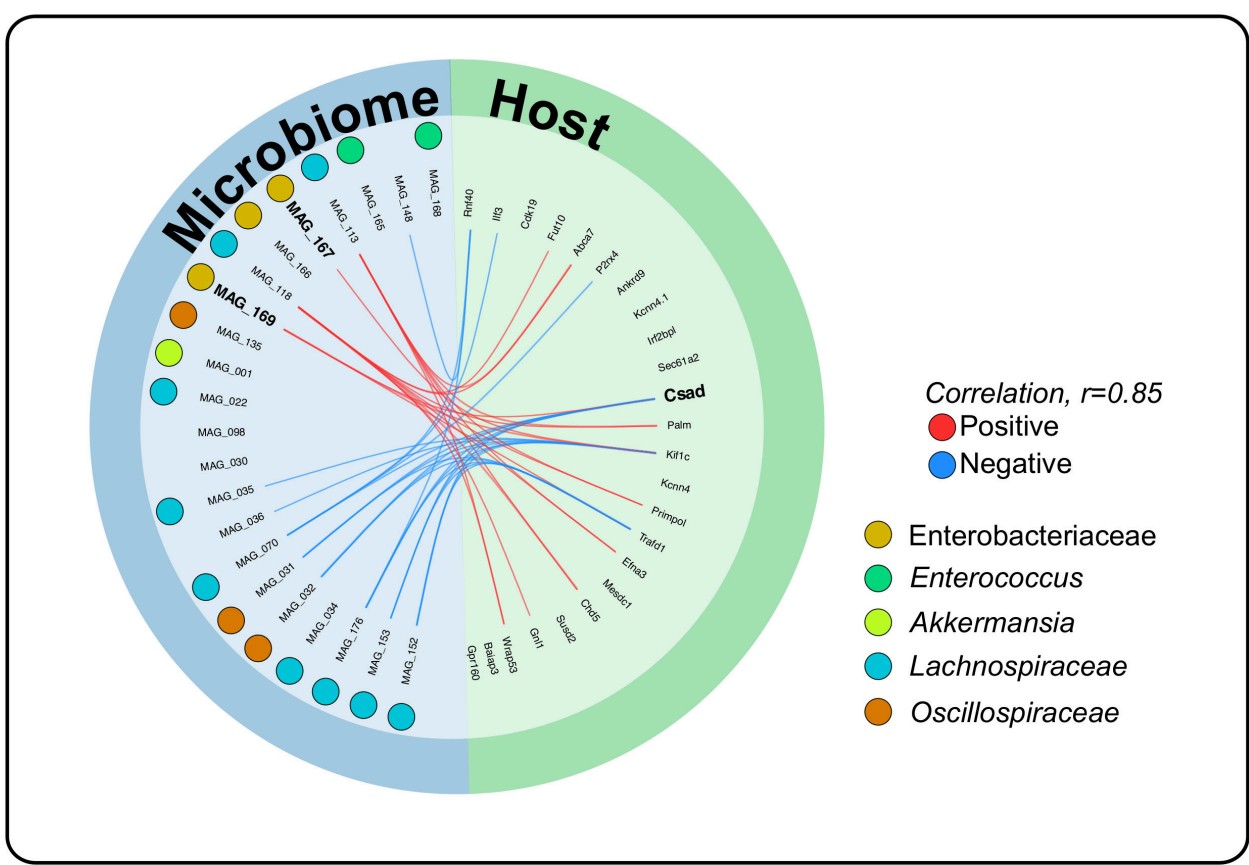

**FIG 5** Coherence between host and microbe highlighted attributes to worsened outcomes in IBD. Correlation analysis of non-redundant MAG detection values with RNA sequencing counts for a subset of host genes showed a strong correlation between MAGs-Enterobacteriaceae and host-expressed CSAD gene. In addition to MAGs resolving to Lachnospiraceae and Oscillospiraceae correlating with the downregulation of the CSAD gene.

pathobionts in the gut (24, 25). The *TRAFD1* gene, responsible for controlling the immune response to pathogens, was positively correlated with Enterobacteriaceae MAG 169, and negatively correlated with multiple MAGs from the Lachnospiraceae and Oscillospiraceae families. Collectively, we were able to identify coherent patterns between the host gene expression and microbial populations and showed potential metabolic pathways these MAGs could utilize to drive host response while fueling the growth and metabolism of these pathobionts.

## Utilization capability of amino acids by bacterial members impacted host IBD outcomes

We used the mixOmics discriminant analysis to show that MAGs-Enterobacteriaceae detection displayed strong positive correlations on host inflammatory gene expression, resulting in persistent intestinal inflammation. Several previous studies have indicated that the development of IBD could be associated with microbial factors including pathways driving host inflammation in addition to sustaining colonization, virulence factors such as secretion systems and effectors, and antibiotic resistance genes (12, 93, 94). Therefore, we assessed potential gene functions in the seven MAGs of interest (MAGs-*Akkermansia*, MAGs-*Enterococcus*, and MAGs-Enterobacteriaceae) to examine whether genome-level alterations related to those functions were observed in our resolved MAGs (Fig. 6A). We observed an absence of virulence genes detected in MAGs-*Akkermansia;* however, multiple antimicrobial resistance genes were present.

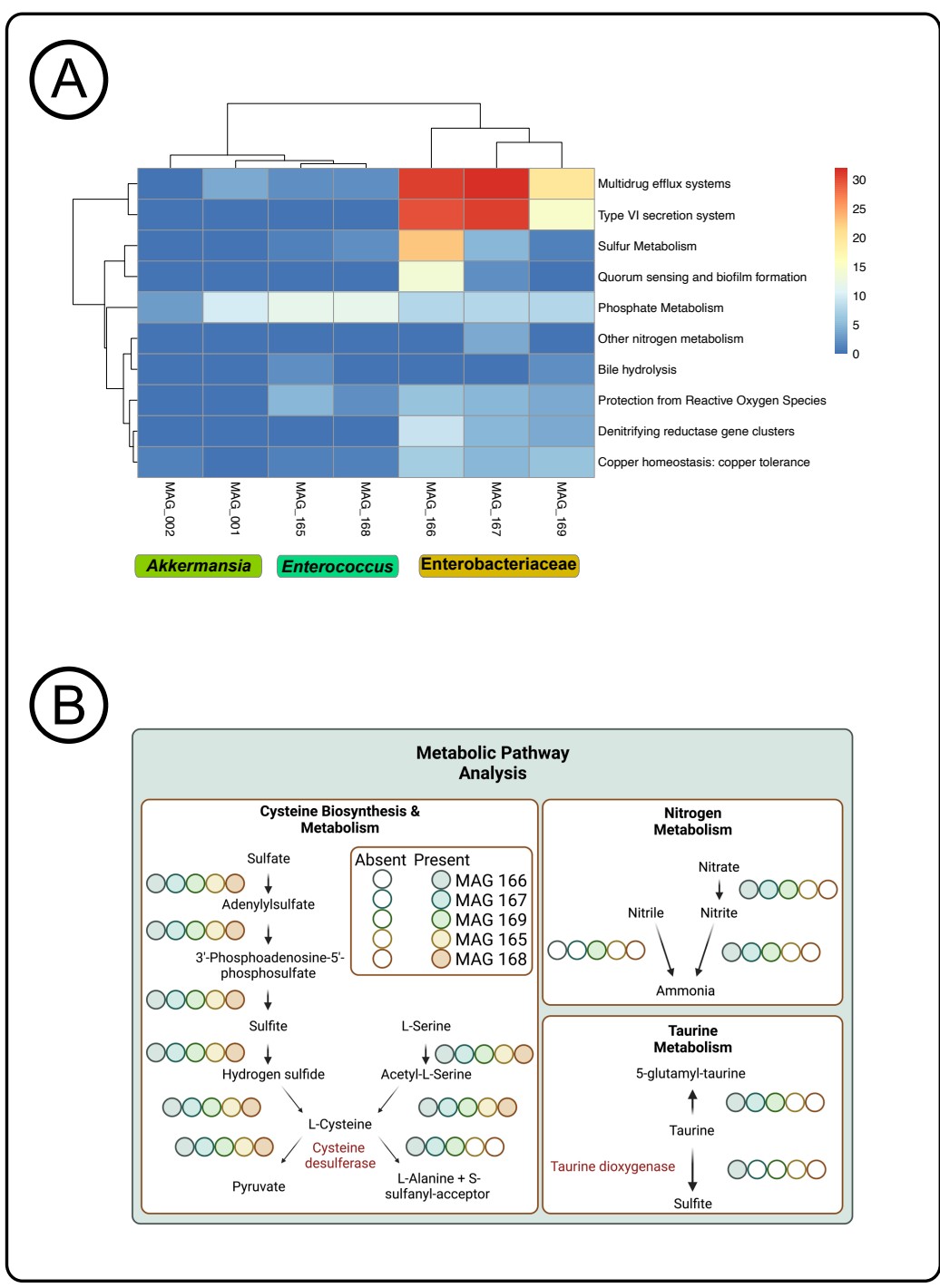

FIG 6  Utilization capability of amino acids by bacterial members impacted host IBD outcomes. (A) Clustering based on subsystem metabolic pathways of each MAG of interest *Enterococcus* and Enterobacteriaceae and *Akkermansia*. (B) Metabolic pathway maps of each MAG of interest *Enterococcus* and Enterobacteriaceae and the genes in sulfur, taurine, and nitrogen metabolisms that are present. Together, this showed the potential of MAGs-Enterobacteriaceae and MAGs-*Enterococcus* to metabolize cysteine, nitrogen, and taurine.

Previously, *Akkermansia muciniphila* has been implicated as a commensal member of the gut microbiota, as a mucus degrader, and has been associated with reduced overall colon inflammation, which may be due to its role in strengthening tight junctions of

intestinal epithelial cells (17, 95, 96). MAGs-Enterobacteriaceae clustered on its own branch with highly detected denitrifying gene clusters, T6SS, and multidrug efflux pumps (Fig. 6A). Our results, along with other studies, found a relative increase in Enterobacteriaceae bacterial populations that display genes that aid in proliferating in the inflamed gut like in IBD patients (19, 97). Our results indicated that the high detection of MAGs-Enterobacteriaceae in dysbiotic mice resulted in more severe intestinal inflammation in these mice, and MAGs-Enterobacteriaceae possessed the potential functions to exacerbate the inflamed gut. In addition, the MAGs-Enterobacteriaceae group included a large presence of nitrate and nitrite reductases capable of denitrifying nitrate and nitrite to ammonia (Fig. 6A). COG genes from both MAGs-Enterobacteriaceae and MAGs-*Enterococcus* groups included the *NarL* and *NarX* gene families, a two-component system regulating various nitrate and nitrite reductases (Table S5). The presence of nitrate and nitrite reductases demonstrated MAGs-Enterobacteriaceae's ability to completely denitrify nitrate to ammonia, which is critical for quick growth under anaerobic conditions like the gut environment. Some of the MAGs in both groups MAGs-*Enterococcus* and MAGs-Enterobacteriaceae possessed sulfur metabolism functional potential (Fig. 6B). The amino acids cysteine and taurine are utilized and broken down during both host and microbial sulfur metabolisms (98, 99). Previous studies have shown the importance of cysteine and taurine in host remediation of intestinal inflammation by providing regulation to pro-inflammatory cytokines and relieving oxidative stress in addition to lumen endothelial cell dysfunction (29, 100).

An important component of the host's inflammatory immune response is the ability of the host to reduce and limit the inflammation. We thus hypothesized that MAGs-*Enterococcus* and MAGs-Enterobacteriaceae might limit the availability of cysteine and taurine to the host, contributing mechanistically to persistent intestinal inflammation. We analyzed the potential metabolic pathways of MAGs-Enterobacteriaceae and MAGs-*Enterococcus* to understand the metabolic activity in these MAGs that coincided and competed with host metabolism. We used the KEGG pathway analysis and showed that MAGs-Enterobacteriaceae possessed complete desulfurization gene pathways, involving both biosynthesis and degradation of the amino acids cysteine and taurine, whereas MAGs-*Enterococcus* possessed complete metabolic pathways for cysteine alone (Fig. 6A). All of the Enterobacteriaceae MAGs harbored two genes; *SufS*, a cysteine desulfurase and *TauD*, a taurine dioxygenase, that are essential for degradation of these two sulfur-containing amino acids by cleaving the sulfur group (101–103). *SufE*, a cysteine desulfurase gene, was identified in MAGs-*Enterococcus* which breaks down cysteine similarly to Enterobacteriaceae (104). To further verify the utilization of cysteine in MAGs-Enterobacteriaceae, we cultured and isolated 31 Enterobacteriaceae bacteria colonies from CPZ/FMT, control/no-CPZ, CPZ/gavage-with-PBS, and CPZ/no-gavage mice fecal samples. Altogether, eight isolates were cultured from mice receiving FMT, eight from the CPZ/gavage-with-PBS group, eight from the CPZ/no-gavage group, and seven from control/no-CPZ mice (Fig. S4). Sanger sequencing revealed that six of the FMT isolates resolved to the genus *Enterobacter*, one to *Klebsiella,* and one to *Pantoea*. Five isolates from the No Gavage group resolved to *Klebsiella*, and two to *Enterobacter*. Most PBS group isolates resolved to *Klebsiella* ($n = 5$), with one *Enterobacter* and one *Kosakonia*. Finally, the control group consisted of five *Klebsiella*, one *Pseudoalteromonas*, and one *Enterobacter*. Overall, these isolates represent mostly *Enterobacter* and *Klebsiella* with several isolates matching the same species (Table S6). We conducted nutrient dependency assays using these 31 bacterial populations with and without cysteine in M9 broth (Table S6). Isolates grown in cysteine-supplemented media resulted in a significantly ($P < 0.05$) higher growth curve than isolates in minimal media indicated at each hour (Fig. 7A). Isolates cultured from the FMT groups showed no significant differences at any time point during the assay, with control isolates differing at the final timepoint (Fig. 7A). Eight isolates from the No Gavage group displayed significantly increased growth rate compared to the unsupplemented M9 broth (Fig. 7A). These results suggest that Enterobacteriaceae isolated from dysbiotic mice were using cysteine more efficiently for growth, whereas

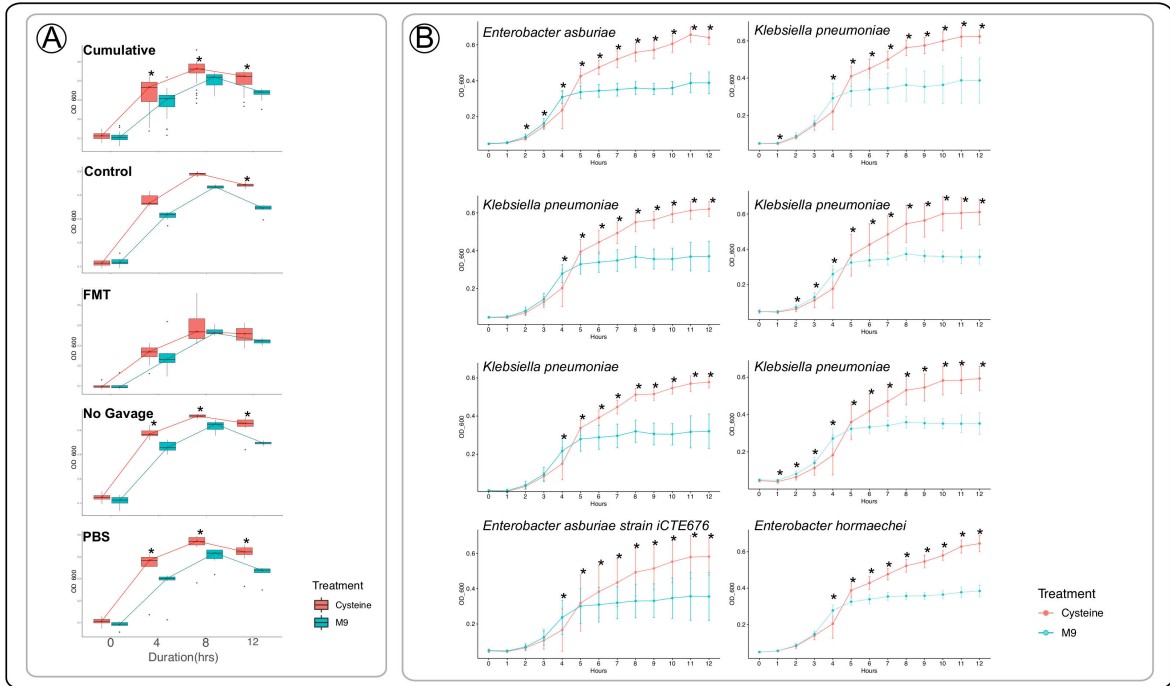

**FIG 7** Nutrient dependency screen of 31 isolates cultured with or without cysteine from IL-10 KO mice in each treatment group. (A) Verifying the differential cysteine utilization capability of Enterobacteriaceae isolates from treatment groups. (B) The growth curve in eight of the dysbiotic group isolates indicating *Enterobacter* and *Klebsiella* strains utilize L-cysteine quickly for increased growth advantage (* indicates $P < 0.05$).

Enterobacteriaceae from mice that received FMT did not indicate this increased growth. This result along with the knowledge that the host experiences low cysteine levels highlights an important potential mechanism that these bacteria could shift to sulfur metabolism to fuel growth and host inflammation, which has been previously speculated (25, 28, 81). To determine which genera may utilize L-cysteine more efficiently in the No Gavage and PBS gavage isolates, eight strains ($n = 5$ *Klebsiella* and $n = 3$ *Enterobacter*) were used for a second nutrient dependency assay performed in duplicate (Fig. 7B). These individual isolates revealed that in addition to using cysteine for significant growth advantage, this advantage began between 4 and 5 hours after inoculation. Our study suggests that this utilization occurs quickly resulting in a significant growth advantage over the absence of L-cysteine supplementation in *Enterobacter* and *Klebsiella* isolates from dysbiotic mice.

## DISCUSSION

We observed the mentioned MAGs were highly detected in the dysbiotic mice with inflamed guts. The identification of cysteine and taurine metabolizing genes in these bacterial populations, alongside the demonstrated use of cysteine for increased growth, suggests that the aforementioned MAGs limit the availability of cysteine to the host. The accessibility of cysteine is essential for restoring gut immune homeostasis by attenuating inflammatory responses (24, 25, 28). Altogether, our results indicated the potential for our resolved MAGs to utilize essential host nutrients for metabolism in addition to utilizing host immune response products like nitrate and nitric oxide for the promotion of growth. These functional mechanisms alongside the potential virulence factors identified provide an advantage for these MAGs to over colonize in the dysbiotic gut, which may contribute to persistent chronic intestinal inflammation in the host, driving disease symptoms and progression.

Here, we use an array of host physiological and genetic data in addition to microbial metagenomic and metabolic information, for the first time, providing insights into possible mechanistic explanations of persistent intestinal inflammation in a dysbiotic gut. We showed a set of potential metabolic pathways that pathobionts are capable of utilizing to gain a foothold in the dysbiotic gut, driving persistent host inflammation, and leading to colitis. We demonstrated in our study that members of Enterobacteriaceae have the functional capabilities to compete with the host for L-cysteine as a sulfur source while driving inflammation with the products of this sulfur metabolism. Another well-known pathway involved with gut inflammation is nitrate metabolism (20, 105). The host produces nitrate and nitric oxide (68, 97) to ward off potential pathogens; however, several Enterobacteriaceae members are tolerant and use one or both of these host response products (106). Essentially, the host inflammatory response fuels the growth of these pathobionts which stimulates further immune modulation. These metabolic pathways that we are investigating could harbor the mechanisms these bacterial populations use to transition from commensal to pathobiont (25). Thus, switching to these alternate inflammation-driving metabolisms creates a chronic cycle of host inflammation and damage to the colon. While our findings provide a compelling link between the Enterobacteriaceae and *Enterococcus* MAGs and sulfur metabolism, these are potential functions resulting from MAGs that were further validated using cultured isolates from the mice in this project by a nutrient dependency assay with L-cysteine indicating the use of these amino acids. We have shown these MAGs were similar to the isolates used for the nutrient dependency assay but are likely not identical. Understanding what drives these alternate metabolisms may prove invaluable for the prevention and therapeutic development of chronic inflammatory disorders like ulcerative colitis and IBD. More research is needed to obtain full genomes of these isolates and compare them to the MAGs mentioned throughout this study, expanding this foundational work for more mechanistic and *in vivo* studies.

## ACKNOWLEDGMENTS

We would like to acknowledge the services and help of Dr. Tracy Meisner for his technique and experimental design. We appreciate Clark Bloomer and the team with the Genome Sequencing Facility at the University of Kansas Medical Center for their help with shotgun metagenomic sequencing. In addition, we would like to thank Dr. Sherry Fleming with the Johnson Cancer Center at Kansas State University for her support and helpful discussion. We also acknowledge the Johnson Cancer Center at Kansas State University for funding and support for this project and personnel. We also thank Dr. Sherry Fleming for her continuous support and advice on this study. Gratitude is also extended to the University of Kansas Medical Center Genome Sequencing Facility for their expertise and assistance in sequencing, including Clark Bloomer, Dr. Veronica Cloud, Rosanne Skinner, and Yafen Niu. Finally, we would like to acknowledge the 52 mice included in this study for their sacrifice to build our understanding of complex diseases such as IBD.

T.G.R. and S.T.M.L. designed the study. Sample collection was performed by T.G.R., L.H., A.K., S.P., K.M., T.S., H.W., and S.S. T.S., T.G.R., L.H., and A.K. completed DNA extraction with Nanodrop and Qubit quality analysis. Q.R. performed RNAseq bioinformatic analysis. T.G.R. and S.T.M.L. performed anvi'o and PATRIC bioinformatic analyses. Bacterial culturing and isolation were done by T.G.R. and S.S. T.G.R. and S.T.M.L. attributed biological relevance, wrote the manuscript, and prepared figures and supplemental files. T.G.R., S.T.M.L., and B.L.P. performed major manuscript and figure refinement, while the remaining authors contributed to lighter refinement. B.L.P. performed histologic analyses. S.T.M.L. acquired funding for this study. All authors read, contributed to the manuscript revision, and approved the submitted version.

This project was supported by an Institutional Development Award (IDeA) from the National Institute of General Medical Sciences of the National Institutes of Health under grant number P20 GM103418. The content is solely the responsibility of the

authors and does not necessarily represent the official views of the National Institute of General Medical Sciences or the National Institutes of Health. We greatly appreciate assistance from the following sources: Kansas IDeA Network of Biomedical Research Excellence (K-INBRE), Kansas State University Johnson Cancer Research Center, Kansas State University Biology Graduate Student Association, Kansas Intellectual and Developmental Disabilities Research Center (NIH U54 HD 090216), the Molecular Regulation of Cell Development and Differentiation—COBRE (P30 GM122731-03)—the NIH S10 High-End Instrumentation Grant (NIH S10OD021743), and the Frontiers CTSA grant (UL1TR002366) at the University of Kansas Medical Center, Kansas City, KS.

The authors disclose no conflicts of interest.

## AUTHOR AFFILIATIONS

[1]Division of Biology, Kansas State University, Manhattan, Kansas, USA

[2]Department of Diagnostic Medicine and Pathobiology, Kansas State University, Manhattan, Kansas, USA

## AUTHOR ORCIDs

Tanner G. Richie  http://orcid.org/0000-0002-2554-6005
Sonny T. M. Lee  http://orcid.org/0000-0001-8073-989X

## FUNDING

| Funder | Grant(s) | Author(s) |
| --- | --- | --- |
| HHS \| NIH \| National Institute of General Medical Sciences (NIGMS) | P20 GM103418 | Sonny T. M. Lee |
| Kansas IDeA Network of Biomedical Research Excellence (K-INBRE) | | Sonny T. M. Lee |
| KSU \| Johnson Cancer Research Center, Kansas State University (JCRC) | | Sonny T. M. Lee |
| Kansas Intellectual and Developmental Disabilities Research Center | NIH U54 HD 090216 | Sonny T. M. Lee |
| Molecular Regulation of Cell Development and Differentiation | P30 GM122731-03 | Sonny T. M. Lee |
| NIH S10 High-end Instrumentation Grant | NIH S10OD021743 | Sonny T. M. Lee |
| KU \| Frontiers Clinical and Translational Science Institute, University of Kansas (KU CTSI) | UL1TR002366 | Sonny T. M. Lee |

## AUTHOR CONTRIBUTIONS

Tanner G. Richie, Conceptualization, Data curation, Formal analysis, Methodology, Project administration, Validation, Visualization, Writing – original draft, Writing – review and editing | Leah Heeren, Data curation, Writing – review and editing | Abigail Kamke, Data curation | Kourtney Monk, Data curation | Sophia Pogranichniy, Data curation | Trey Summers, Data curation, Writing – review and editing | Hallie Wiechman, Data curation | Qinghong Ran, Data curation, Formal analysis, Methodology, Software, Writing – original draft, Writing – review and editing | Soumyadev Sarkar, Data curation, Formal analysis, Writing – original draft, Writing – review and editing | Brandon L. Plattner, Data curation, Formal analysis, Validation | Sonny T. M. Lee, Conceptualization, Data curation, Formal analysis, Funding acquisition, Investigation, Methodology, Project administration, Resources, Software, Supervision, Validation, Visualization, Writing – original draft, Writing – review and editing

## DATA AVAILABILITY

Raw sequencing data for the mice metagenomes were uploaded to SRA under NCBI BioProject PRJNA843079. All other analyzed data, including supplemental figures and tables as well as data in the form of databases and fasta files are accessible at figshare https://doi.org/10.6084/m9.figshare.21753623.v2.

## ETHICS APPROVAL

All mouse experiments were reviewed and approved by the Institutional Animal Care and Use Committee at Kansas State University (APN #4391).

## ADDITIONAL FILES

The following material is available online.

### Supplemental Material

**Figure S1 (mSystems00703-23-S0001.pdf).** Clustered heatmap of all mice with MAG detection ratios.
**Figure S2 (mSystems00703-23-S0002.pdf).** Gene ontology comparisons of each mouse treatment group.
**Figure S3 (mSystems00703-23-S0003.pdf).** Phylogenetic trees of the 7 MAGs of interest.
**Figure S4 (mSystems00703-23-S0004.pdf).** Nutrient dependency assay results for each isolate.
**Table S1 (mSystems00703-23-S0005.xlsx).** Metagenomic result tables.
**Table S2 (mSystems00703-23-S0006.xlsx).** Mouse pup weights, histology, and cytokine results.
**Table S3 (mSystems00703-23-S0007.xlsx).** RNA sequencing Deseq data.
**Table S4 (mSystems00703-23-S0008.xlsx).** MAG detection ratio matrix.
**Table S5 (mSystems00703-23-S0009.xlsx).** MAGs of interest gene annotation.
**Table S6 (mSystems00703-23-S0010.xlsx).** Nutrient dependency assay isolate IDs.

### Open Peer Review

**PEER REVIEW HISTORY (review-history.pdf).** An accounting of the reviewer comments and feedback.

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
