## [Reviewer comments · mSystems]

Limitation of Amino Acid Availability by Bacterial Populations During Enhanced Colitis in IBD Mouse Model

Tanner Richie, Leah Heeren, Abigail Kamke, Sophia Pogranichniy, Kourtney Monk, Trey Summers, Hallie Wiechman, Qinghong Ran, Soumyadev Sarkar, Brandon Plattner, and Sonny Lee

Corresponding Author(s): Sonny Lee, Kansas State University

Review Timeline:

Submission Date:	July 10, 2023
Editorial Decision:	September 18, 2023
Revision Received:	September 19, 2023
Accepted:	September 27, 2023

Editor: Barbara Methe

Reviewer(s): The reviewers have opted to remain anonymous.

Transaction Report:

DOI: <https://doi.org/10.1128/msystems.00703-23>

September 18, 2023

Dr. Sonny T.M. Lee
Kansas State University
Division of Biology
Manhattan

Re: mSystems00703-23 (Limitation of Amino Acid Availability by Bacterial Populations During Enhanced Colitis in IBD Mouse Model)

Dear Dr. Sonny T.M. Lee:

Thank you for submitting your manuscript to mSystems. We have completed our review and I am pleased to inform you that, in principle, we expect to accept it for publication in mSystems. However, acceptance will not be final until you have adequately addressed the reviewer comments. Please pay close attention to all remaining comments from both reviewers as they are relevant to clarifying the work presented in this manuscript.

Preparing Revision Guidelines

Please return the manuscript within 60 days; if you cannot complete the modification within this time period, please contact me. If you do not wish to modify the manuscript and prefer to submit it to another journal, please notify me of your decision immediately so that the manuscript may be formally withdrawn from consideration by mSystems.

Sincerely,

Barbara Methe

Editor, mSystems

Journals Department
Reviewer comments:

Reviewer #1 (Comments for the Author):

General comments:

The resubmitted work shows significant quality improvements over the previous version in each major section of the manuscript. It also solves the shortcomings detected in some of the figures. Last but not least, the raw data is now public, which is the basis for any replication attempts by the research community or use in any metastudy. However, there are still a couple of minor issues that have not been fully addressed by the authors. See below for specific comments.

Specific comments:

1) My comment was:

*** 15. Line 118: Please don't write so many significant figures as most of them are meaningless. "12.2 {plus minus} 1.6 million sequences" should be more than enough. The value after the plus-minus sign may be understood as an estimation of the error or uncertainty of the previous value, but is that your intention? For instance, is that the error of the mean or a measure of the dispersion of the sample?

Authors' reply is:

>>> We have trimmed or eliminated the use of so many significant figures for conciseness. "average of 12,228,040 {plus minus} 1,646,545 reads per metagenome" now reads "average of 12.2 million {plus minus} 1.6 million reads per metagenome". (PAGE 11, LINE 265). This was also addressed throughout the results section, specifically with the detection values.

• Comment (Line 267 of the resubmitted manuscript): Thanks for changing the numerical representation to follow the standards about significant figures, but my questions were not answered. Please clarify in the text, as it's the first but not the single appearance of this construction, whether the value after the plus-minus sign (in this and other occasions) is the standard error of the mean.

2) My comment was:

*** 29. Lines 275 to 286: There are a lot of issues with numerical representations in this long paragraph: please correct numbers with too many figures clearing non-significative ones and correct errors with more precision than the precision of the magnitude.

Authors' reply is:

>>> This area of the text has been modified for simplicity and easy readability. It read "(control/no-CPZ: 0.894 {plus minus} 0.0496; CPZ/no-gavage: 0.997 {plus minus} 0.0005; CPZ/gavage-with-PBS: 0.924 {plus minus} 0.0505; CPZ/FMT: 0.998 {plus minus} 0.0005), regardless of whether the mice were dysbiotic or not. Although control mice had significantly lower MAG 001 detection compared with other groups ($p < 0.0421$, Figure 4B)," and now reads "(control/no-CPZ: 0.89; CPZ/no-gavage: 0.99; CPZ/gavage-with-PBS: 0.92; CPZ/FMT: 0.99)," with the error removed. (PAGE 17, LINE 410)

• Comment (Page 18 of the resubmitted manuscript): Thanks for improving the readability of the entire section. However, my comment was about correcting the numerical representation regarding the number of significant digits of each magnitude with its uncertainty, not to keep the former and remove the latter. Finally, the lines 429 and 432 contain the construction " $p = \{space\}$ "

3) My comment was:

>>> Line 495: As general comments about the "Metagenomics" subsection, (i) details about the different databases used are missing (e.g., date of build, size), and they are as important as the details about the codes used; (ii) I miss some discussion about possible contamination sources, how they may affect the analysis, and the procedures and techniques that have been used to eliminate or mitigate them (as an example, the use of end-to-end negative control samples for this purpose).

Authors' reply is:

*** We have clarified the "Metagenomics" section to point out to the readers clearly that all versions of the databases used are detailed in following paragraphs in the "Metagenomics" subsection. The sentence now reads "The workflows use Snakemake (37), and implemented numerous tasks detailed in the following sections," (PAGE 7, LINE 182)

We added the following line in the Methods section "A negative-control sample was also included in the process to ensure there was no contamination from molecular DNA extraction and sequencing." to improve clarity for the readers. (PAGE 7, LINE 161)

We also added the following in the Discussion section "While our findings provide a compelling link between the Enterobacteriaceae and Enterococcus MAGs and sulfur metabolism, these are potential functions resulting from MAGs that were further validated using cultured isolates from the mice in this project by a nutrient dependency assay with L-cysteine indicating the use of these amino acids. We have shown these MAGs were similar to the isolates used for the nutrient dependency assay, but are likely not identical." to show the limitations in our study. (PAGE 24, LINE 567)

- Comment (Bioinformatics section on the resubmitted manuscript): thank you very much for addressing the point (ii) of my comment. However, despite authors' claims, the point (i) has not been addressed: details of the databases (DBs) used are still missing. The details about the DBs are as important as the details about the software used. Both software and data should be detailed as requirement for any attempt of replication.

Reviewer #2 (Comments for the Author):

Thank you to the authors for their careful responses. The authors have addressed the majority of my concerns and the new growth curve and sanger sequencing results are helpful. However, I continue to find that certain areas of the manuscript 'over-conclude', and would therefore change the phrasing in the following areas:

1. Abstract lines 32-33 and importance lines 40-41. The manuscript never directly shows that Enterobacteriaceae metabolism drives host inflammation. (To do so would have required an intervention in the mice with specific strains and mutants). Please change the phrasing to emphasize this is only a likely mechanism based on association, rather than a definite conclusion from this paper. E.g. replace 'driving' with 'associated with'.
2. Line 369-370 conclusion here does not fit results re: cytokines; FMT mice did not display less host-associated cytokine modulation, they displayed the same or more. Please re-phrase the conclusion to make it clear that only histological inflammation, not cytokines, decreased.
3. line 732-734. This conclusion is misleading since isolates from healthy mice display similar cysteine-based growth boost as isolates from dysbiotic mice. Also, there was never a direct statistical comparison between FMT isolates + cysteine and dysbiotic isolates + cysteine. It would be more accurate to say that "Enterobacteriaceae from dysbiotic mice use cysteine highly efficiently for growth", without any attempt at comparison between groups.

Dear Dr. Methe,

We are very thankful for the helpful suggestions and comments to further improve this manuscript. These suggestions were incorporated in the revised version of the manuscript with our responses included (colored blue text). In brief, we:

- Addressed all statistical variances in the manuscript as standard error of the mean, and replaced all variances in the manuscript that fit the significant figures.
- Improved the clarity of several areas including the Abstract, serum data and growth curve results.
- Checked for accuracy of all programs and databases used in this study to ensure they are included in the “Materials and Methods” section, and added all versions used.

In our revision we have included a revised “CLEAN” and “MARKED-UP” version for transparency of the edits made. All page and line numbers will correspond with the “CLEAN” version of the manuscript. We hope that we have clarified the remaining comments the reviewers brought up and thank you again for the improvements that you and the reviewers have offered.

Reviewers Comments:

Reviewer #1

1. Comment (Line 267 of the resubmitted manuscript): Thanks for changing the numerical representation to follow the standards about significant figures, but my questions were not answered. Please clarify in the text, as it's the first but not the single appearance of this construction, whether the value after the plus-minus sign (in this and other occasions) is the standard error of the mean.

Thank you to the reviewer for the thorough clarification. We have added the term "mean \pm SE" in both the "Methods" and "Results" sections in the first appearance for better clarity. We have noted this in the manuscript as follows:

(Page 11 Line 265)
"Shotgun sequencing of the fecal samples resulted in a total of over 623 million sequences with an average of 12.2 ± 1.6 (mean \pm SE) million reads per metagenome."

(Page 10 Line 234)
as well as mentioning this in detail under the "Statistical analyses" section of the "Materials and Methods" "P-values of less than 0.05 were considered statistically significant, with standard error of the mean (mean \pm SE) utilized where appropriate."

2. Comment (Page 18 of the resubmitted manuscript): Thanks for improving the readability of the entire section. **However, my comment was about correcting the numerical representation regarding the number of significant digits of each magnitude with its uncertainty, not to keep the former and remove the latter.** Finally, the lines 429 and 432 contain the construction "p = {space}<number" several times. It's unclear to me if the intended meaning is "equal or less than" (then the space should be moved to the right, between the sign and the number), or the equal sign was included by mistake.

We thank the reviewer for clarification in this section. The magnitude of uncertainty (SE) was added back in the correct lines using fewer significant figures for the values (mean) matching the magnitude of the error. It is shown in bold in this reply.

(Page 18 Lines 412-421)
" Out of the three groups of MAGs of interest, MAGs-*Akkermansia* was highly detected in all of the samples (control/no-CPZ: **0.89 ± 0.0496** ; CPZ/no-gavage: **0.99 ± 0.0005** ; CPZ/gavage-with-PBS: **0.92 ± 0.0505** ; CPZ/FMT: **0.998 ± 0.0005**), regardless of whether the mice were dysbiotic or not. MAGs-*Enterococcus* and MAGs-Enterobacteriaceae had a very different detection pattern as that of MAGs-*Akkermansia*

(Figure 4B). Both MAGs-*Enterococcus* and MAGs-Enterobacteriaceae displayed low detection in the control/no-CPZ (MAGs-*Enterococcus*: **0.005 ± 0.0015**, MAGs-Enterobacteriaceae: **0.005 ± 0.0035**) and CPZ/FMT (MAGs-*Enterococcus*: **0.002 ± 0.0005**, MAGs-Enterobacteriaceae: **0.003 ± 0.0006**) groups, however, were highly detected in the CPZ/no-gavage (MAGs-*Enterococcus*: **0.08 ± 0.0501**, MAGs-Enterobacteriaceae: **0.05 ± 0.0226**) and CPZ/gavage-with-PBS (MAGs-*Enterococcus*: **0.34 ± 0.0776**, MAGs-Enterobacteriaceae: **0.30 ± 0.0583**) group..”

(Page 18 Lines 422 and 430) were updated to remove the equal sign as this was not necessary and the meaning was p less than, not equal to.

“(MAG 166: p = <0.0001; MAG 167: p = <0.011; MAG 169: p = <0.0005), however, not in the no gavage (MAG 166: p = 0.39; MAG 167: p = 0.94; MAG 169: p = 0.052). MAGs-*Enterococcus* showed a similar trend of control mice and FMT mice showing lower detection ratios than PBS gavage (MAG 165: p = <0.0001; MAG 168: p = 0.068)”

Now reads: “Detections of Enterobacteriaceae MAGs by treatment group indicated a lower presence in control/no-CPZ and mice that receive FMT with no significant differences between these groups, however significantly higher detections were observed in the dysbiotic mice with the PBS gavage (MAG 166: p < 0.0001; MAG 167: p < 0.011; MAG 169: p < 0.0005), however, not in the no gavage (MAG 166: p = 0.39; MAG 167: p = 0.94; MAG 169: p = 0.052). MAGs-*Enterococcus* showed a similar trend of control mice and FMT mice showing lower detection ratios than PBS gavage (MAG 165: p < 0.0001; MAG 168: p = 0.068) however no difference in the no gavage group (MAG 165: p = 0.42 MAG 168: p = 0.99) respectively (Figure 4B).”

3. Comment (Bioinformatics section on the resubmitted manuscript): thank you very much for addressing the point (ii) of my comment. However, despite authors' claims, the point (i) has not been addressed: details of the databases (DBs) used are still missing. The details about the DBs are as important as the details about the software used. Both software and data should be detailed as requirement for any attempt of replication.

(Page 7 Line 179)

Please find the list below of databases and programs utilized for this study taken directly from the methods section. Please note that “anvi-gen-contigs-database” utilizes the contig database we assembled from the metagenome short-reads in this study using MEGAHIT, and is available in the figshare file <https://doi.org/10.6084/m9.figshare.21753623.v2>. This database represents all contigs that were assembled which we then mapped our metagenomes to resolve our MAGs (full details in methods of the manuscript).

- anvi'o version 7.1 - which includes the following bioinformatical steps:
 - anvi-run-workflow
 - anvi-gen-contigs-database
 - anvi-run-hmms
 - anvi-run-ncbi-cogs - COG version 2014
 - anvi-run-kegg-kofams - KOfams version 2021-10-03, KEGG version 100.0
 - anvi-profile
 - anvi-merge
 - anvi-cluster-contigs
 - anvi-refine
 - anvi-compute-genome-similarity
- Snakemake 6.15.5
- Hidden Markov Model (HMM) ver 11.0
- lu-filter-quality-minoche
- MEGAHIT v1.2.9
- Prodigal v2.6.3
- HMMER v 3.2.1
- Bowtie2 v 2.3.5
- Samtools v1.9
- CONCOCT v1.1.0
- PyANI v0.2.9

Reviewer #2 (Comments for the Author):

Thank you to the authors for their careful responses. The authors have addressed the majority of my concerns and the new growth curve and sanger sequencing results are helpful. However, I continue to find that certain areas of the manuscript 'over-conclude', and would therefore change the phrasing in the following areas:

1. Abstract lines 32-33 and importance lines 40-41. The manuscript never directly shows that Enterobacteriaceae metabolism drives host inflammation. (To do so would have required an intervention in the mice with specific strains and mutants). Please change the phrasing to emphasize this is only a likely mechanism based on association, rather than a definite conclusion from this paper. E.g. replace 'driving' with 'associated with'.

We thank the reviewer for the helpful suggestions to the Abstract and Importance sections. To improve clarity for the readers, we mentioned multiple times in the revised manuscript, that these are potential pathways. We have made the suggested edits to the sections from:

“Our results show that microbial populations use alternate metabolisms and sequester host nutrients for growth, driving inflammation in the gut.

Importance

Inflammatory bowel disease is associated with an increase in Enterobacteriaceae and *Enterococcus* species, however, the specific mechanisms are unclear. Previous research has reported associations of the microbiota and inflammation, here we investigate potential pathways specific bacteria populations use to drive gut inflammation. Richie *et al.* shows that these bacterial populations utilize an alternate sulfur metabolism and are tolerant of host-derived immune-response products. These metabolic pathways drive host gut inflammation and fuels over colonization of these pathobionts in the dysbiotic colon. Cultured isolates from dysbiotic mice indicated faster growth supplemented with L-cysteine, showing these microbes can utilize essential host nutrients.”

(Line 14-39)

To now read:

Abstract

“Our results show that microbial populations use alternate metabolisms and sequester host nutrients for growth, associated with inflammation in the gut.”

Importance

“Inflammatory bowel disease is associated with an increase in Enterobacteriaceae and *Enterococcus* species, however, specific mechanisms are unclear. Previous research has reports associations of the microbiota and inflammation, here we investigate potential pathways specific bacteria populations may use to drive gut inflammation. Richie *et al.* shows that these bacterial populations utilize an alternate sulfur metabolism and are tolerant of host-derived immune-response products. These metabolic pathways are associated with host gut inflammation and over colonization of these pathobionts in the dysbiotic colon. Cultured isolates from dysbiotic mice indicated faster growth supplemented with L-cysteine, showing these microbes can utilize essential host nutrients.”

2. Line 369-370 conclusion here does not fit results re: cytokines; FMT mice did not display less host-associated cytokine modulation, they displayed the same or more. Please re-phrase the conclusion to make it clear that only histological inflammation, not cytokines, decreased.

We thank the reviewer for catching this point in our summation. The FMT mice had a similar cytokine profile, which was made clear in the text, and to clarify we have adjusted the summary statement to better reflect the data presented above from;

“In summary, IL-10 KO mice receiving FMT displayed less colonic inflammation and host-associated cytokine modulation, indicating a successful FMT.”

(Page 13 Line 300)

To now read:

“In summary, IL-10 KO mice receiving FMT displayed less colonic inflammation with little change in host-associated cytokine modulation compared with other groups, indicating an overall successful FMT.”

3. Line 732-734. This conclusion is misleading since isolates from healthy mice display similar cysteine-based growth boost as isolates from dysbiotic mice. Also, there was never a direct statistical comparison between FMT isolates + cysteine and dysbiotic isolates + cysteine. It would be more accurate to say that "Enterobacteriaceae from dysbiotic mice use cysteine highly efficiently for growth", without any attempt at comparison between groups.

We would like to thank the reviewer for bringing attention to this statement. We have revised this statement to conclude that the isolates from FMT mice did not appear to utilize L-cysteine for growth, but all other groups showing statistically significant growth increases when supplemented.

This statement reads “These results suggest that Enterobacteriaceae in these dysbiotic mice were using cysteine more efficiently for growth than Enterobacteriaceae from healthy mice or mice that received FMT..”

(Page 23 Line 528)

and now reads “These results suggest that Enterobacteriaceae isolated from dysbiotic mice were using cysteine more efficiently for growth, whereas Enterobacteriaceae from mice that received FMT did not indicate this increased growth.”

September 27, 2023

Dr. Sonny T.M. Lee
Kansas State University
Division of Biology
Manhattan

Re: mSystems00703-23R1 (Limitation of Amino Acid Availability by Bacterial Populations During Enhanced Colitis in IBD Mouse Model)

Dear Dr. Sonny T.M. Lee:

Thank you for completing the additional changes as recommended by the reviewers.

Your manuscript has been accepted, and I am forwarding it to the ASM Journals Department for publication. For your reference, ASM Journals' address is given below. Before it can be scheduled for publication, your manuscript will be checked by the mSystems production staff to make sure that all elements meet the technical requirements for publication. They will contact you if anything needs to be revised before copyediting and production can begin. Otherwise, you will be notified when your proofs are ready to be viewed.

If you would like to submit a potential Featured Image, please email a file and a short legend to msystems@asmusa.org. Please note that we can only consider images that (i) the authors created or own and (ii) have not been previously published. By submitting, you agree that the image can be used under the same terms as the published article. File requirements: square dimensions (4" x 4"), 300 dpi resolution, RGB colorspace, TIF file format.

We recognize that the video files can become quite large, and so to avoid quality loss ASM suggests sending the video file via <https://www.wetransfer.com/>. When you have a final version of the video and the still ready to share, please send it to mSystems staff at msystems@asmusa.org.

Sincerely,

Barbara Methe
Editor, mSystems

Journals Department
E-mail: mSystems@asmusa.org